# Peer review of "Unveiling the Potential: Remote Monitoring and Telemedicine in Shaping the Future of Heart Failure Management"

_life, 2024, doi:10.3390/life14080936_

Round 1

Reviewer 1 Report

Comments and Suggestions for Authors

This is a narrative review of the role of telemedicine in the monitoring of heart failure; however, the following corrections should be made:

The tables should be clearer: separating each section into columns as they are sometimes difficult to read.

A flow chart explaining the advances in HF telemonitoring in the last 20 years is highly recommended.

A "Limitations" section should be included, explaining that, since it is not a systematic review, it presents weaknesses and selection biases that are difficult to modify inherent to this type of review.

On the other hand, a heading should be included explaining the disadvantages or difficulties of using telemonitoring: for example, in patients with hearing or visual impairments, or with a low cultural/intellectual level. This brief letter serves as an example: 

López Reboiro ML, Sardiña González C, Castro-Conde BA, López Castro J. Telemedicine Yes, But No Telepathy. Acta Med Port. 2020 Sep 1;33(9):624. 

With these changes it could be accepted for publication.

Author Response

Comments 1: The tables should be clearer: separating each section into columns as they are sometimes difficult to read.

Response 1: Thank you for pointing this out. We agree with this comment. Therefore, we have changed the tables.

Comments 2: A "Limitations" section should be included, explaining that, since it is not a systematic review, it presents weaknesses and selection biases that are difficult to modify inherent to this type of review.

Response 2: Thank you for your comments. We have added limitation sentences  「There are some limitations. Given that this is not a systematic review, it inherently possesses certain weaknesses and selection biases that are challenging to mitigate. Additionally, remote monitoring and telemedicine prove to be less practical for patients with hearing or visual impairments or those with lower cultural or intellectual levels. These limitations underscore the need for tailored approaches to ensure equitable access and effectiveness in digital health interventions. 」to emphasize this point in line 319~324.

Reviewer 2 Report

Comments and Suggestions for Authors

thanks for the opportunity to review this interesting paper. I think the scientific value is there although I am a little unsure as to what type of research this is. it seems to me that the work is based on secondary data analysis which is fine, however, authors must state that in their title and somehow point out this as a method of data collection. This could also be a perspective on remote monitoring and telemedicine in HF etc. 

The paper is well written and flows nicely from title to conclusion, however, I found that the authors had multiple sub-conclusions for various sections (example, lines, 123, 156, 215, 258 etc.). I suggest that they leave any summary to the final conclusion of the paper and only address each section based on findings and summary of analysis. 

Table 1- please add any description to the text citing the table and not to the bottom of the table- example: lines 171-172

Author Response

Comments : Table 1- please add any description to the text citing the table and not to the bottom of the table- example: lines 171-172

Response : Thank you for your comments. We have added the following sentences  「Remote monitoring and telemedicine in the management of heart failure present a range of significant benefits and notable challenges. The primary benefits encompass improved patient outcomes through continuous health monitoring, which enables timely interventions and optimal management of the condition. Furthermore, telemedicine enhances patient engagement by providing convenient access to healthcare providers, thereby encouraging active participation in their own care. This approach also contributes to a reduction in hospital readmissions by allowing for early detection of symptoms and prompt management. Additionally, the cost-effectiveness of telemedicine is evident as it reduces the need for in-person visits and hospital stays, ultimately lowering healthcare costs. Increased accessibility is another critical advantage, as telemedicine bridges the gap for patients residing in remote or underserved areas, ensuring they receive the necessary care.

Despite these benefits, several challenges persist. Technical issues such as the dependence on reliable internet connectivity and advanced technology can be significant barriers, particularly in rural areas with poor connectivity. Data privacy and security concerns are paramount, as handling sensitive patient information requires robust security measures to prevent breaches and maintain confidentiality. Patient compliance with using the technology and adhering to medical advice is essential for successful remote monitoring. Health literacy poses another challenge, especially among the elderly, who may struggle with the technology required for telemedicine. Finally, the integration of telemedicine platforms with existing electronic health record systems can be complex and re-source-intensive, posing an additional hurdle to widespread adoption. 」to emphasize this point in line 168~190.

Reviewer 3 Report

Comments and Suggestions for Authors

This is a very nice overview paper about current state, benefits and challenges with health remote monitoring. The paper is proficiently written with very nice flow and high language quality. The paper does not reference all the relevant previous publications, but then again this is not systematic review paper.

The paper presents state and challenges of remote monitoring without going into technical details. Therefore it is best suited for medical personnel and less interesting for technically educated people working in the field of remote monitoring.

Minor comments:

1. Second paragraph in II is a repetition of a previous paragraph.

2. I am not sure if AI and ML are two distinctive disciplines. I am under impression that people mostly think of ML as a part of AI.

Author Response

Comments : Second paragraph in II is a repetition of a previous paragraph.

Response : Thank you for pointing this out. We agree with this comment. Therefore, we have changed first paragraph to avoid repetition. We delete the sentences 「The evolution of remote monitoring and telemedicine, driven by technological advancements and a growing recognition of their potential in enhancing healthcare delivery, especially in chronic disease management like HF, has been remarkable.」in line 92~94.